# A Composite Flexible Sensor for Direct Ventricular Assist Device

**DOI:** 10.3390/s22072607

**Published:** 2022-03-29

**Authors:** Zhong Yun, Kuibing Li, Hao Jiang, Xiaoyan Tang

**Affiliations:** The College of Mechanical and Electrical Engineering, Central South University, Changsha 410083, China; yunzhong@csu.edu.cn (Z.Y.); 193711071@csu.edu.cn (H.J.); txy@csu.edu.cn (X.T.)

**Keywords:** nanostructured materials, flexible electronics, force sensors, pressure sensors

## Abstract

A direct ventricular assist device is one of the effective means to treat patients with heart failure; the key point of the problem is the flexible sensor that can measure the drive pressure and shape variable of the heart auxiliary device. This study was based on the high-voltage electric field guidance process and the porous foaming process, and designed an implantable resistance/capacitive composite flexible sensor that can effectively detect the pressure and deformation signal caused by fine surface contact and pneumatic muscle expansion. Experiments showed the performance of composite sensors with special structure design was greatly improved compared with the control group—the strain measurement sensitivity was 22, pressure measurement sensitivity was up to 0.19 Kpa^−1^. Stable strain measurements were made up to 35 times and pressure measurements over 100 times. In addition, we solved the interference problem of resistance/capacitance flexible sensors through an optimized common substrate process. Finally, we tested a pneumatic muscle direct ventricular assist device with a composite flexible sensor on a model heart; the experiment showed that this resistance/capacitive composite flexible sensor can effectively detect surface contact with pneumatic muscle and the displacement signals.

## 1. Introduction

As a common major disease in the elderly, heart disease will eventually lead to heart failure [1]. For the treatment of left ventricular failure, there are usually the following treatment methods: drug treatment, intraaortic balloon counter-pacing (IABP), mechanical ventricular assist, skeletal muscle ventricular assistance, artificial heart, heart transplantation and others [2]. An artificial heart pump is divided into a pulsating diaphragm pump, an impeller pump and a maglev blood pump [3,4], but such devices usually cause damage to the native heart and blood cells, causing thrombosis, solute blood and other problems. Therefore, studies on direct cardiac auxiliary devices without direct contact with blood are increasing [5].

The direct ventricular assist device is attached to the external wall of the heart and auxiliary supplies blood to the heart by regular compression. In 1965, Anstadt et al. [6] designed a cup-shaped epicardial extrusion device for CARDIopulmonary resuscitation; the outer shell is semi-rigid and the inner layer is a silicone diaphragm, which is connected by a catheter to an air pump that drives the device to assist the heart to compression. In 2017, Roche et al. [7] designed Soft Robotic Sleeve, which controls artificial muscle contraction and relaxation through an air pump to restore the motion state of the heart; experimental results showed that it restored cardiac output to about 97% of normal baseline levels in pigs. EHAM [8] is a teniform direct ventricular assist device; its plastic shell and polyurethane film are connected to its driving part, so as to squeeze the ventricle or increase ventricular filling through the transformation of positive and negative pressure. Our team also studied a direct ventricular assist device based on pneumatic muscle (Figure 1). The device is all composed of silicone that is flexible and biocompatible, and can control the pneumatic muscle expansion and contraction through the air pump to achieve the extrusion of the heart to assist abnormal heart beating.

At present, the structural design and control system [9] has produced some excellent experimental results, effectively improving the blood dissolution and infection problem of the blood pump. However, there are still some shortcomings, for example, the device cannot fully fit with the surface of the heart and is unable to adapt to the air pressure control; research on flexible sensors suitable for the heart auxiliary device is the key to solving the above problems.

The flexible sensor adopts deformable material as the base and can detect deformation and pressure on the curved surface. In principle, it can be divided into piezoresistive, capacitive and piezoelectric sensors [10]. Piezoresistive sensors convert external pressure into changes in resistance signals. In recent years, researchers have developed high-performance piezoresistive sensors by introducing graphene [11,12] and carbon nanotubes [13] into sensitive components. A capacitive flexible sensor can change the distance between poles and dielectric constant under the influence of external stress, and obtain the pressure value by detecting the capacitance change. Piezoelectric sensors are based on the piezoelectric effect, and the external force causes the piezoelectric material to generate an electric charge, thus detecting the magnitude of the force. For example, Persano et al. [14] developed a flexible piezoelectric sensor using the electrochemical deposition method of a polyvinylidene fluoride-trifluoroethylene nanofiber array, which has the advantages of high sensitivity and short response time. Chen et al. [15] proposed a novel nanowire/graphene heterostructure piezoelectric sensor, which can measure high static pressure and short response time to 57 milliseconds.

An external force changes the resistance value of the sensitive element inside the flexible resistance sensor and measures the value of the external force by measuring the resistance, thus indirectly measuring the displacement. The conductive PDMS materials based on carbon nanomaterials have a stable mechanical performance and can withstand large strain, and are one of the preferred sensitive materials for making flexible strain sensors. For example, Liu et al. [16] of the University of Oklahoma combined PDMS and conductive carbon nanoparticles to print lines on the PDMS layer with 3D printing technology, then heated the conductivity pathway, and finally covered PDMS curing to make the flexible strain sensor. The sensor was still characterized by high sensitivity, low hysteresis and good linearity under long force cycles. Zhu et al. [17] used indium tin oxide/polyethylene terephthalate film to produce a microstructured electrode with a fast response time of 0.2 s. Joseph et al. [13] mixed carbon nanoparticles with silicone oil into a liquid conductive material, and injected the material into a flexible substrate with 3D printing technology to produce a flexible tension sensor. Choi. D.Y et al. [18] also added EG/NaCl ion solution as a conductive material into the microchannels of the flexible matrix to prepare a flexible tension sensor with high strain and low hysteresis. Guo et al. [19] of Central South University combined silicon oil with carbon nanotubes into PDMS material to produce high sensitivity conductive PDMS material, from which the flexible resistance sensor could reach 30 times the sensitivity of the ordinary material sensor. Monti et al. [20] prepared conducting PDMS materials with high intensity and high conductivity; however, sensors using such conducting PDMS materials have not been studied.

Pressure sensors convert mechanical pressure signals into electrical signals, which can be roughly divided into capacitive or resistive pressure sensors. For example, Kim et al. [21] inserted the carbon nanotube into polydimethylsiloxane to make a wearable flexible resistance pressure sensor that can be used to detect human joint movement. Wang et al. [22] developed a resistive pressure sensor consisting of two layers of single-wall carbon nanotubes and PDMS films. Due to the microstructure at the substrate, the sensitivity of the sensor was rapidly increased when the applied pressure was less than 300 Pa, which can detect very small pressures. Mannsfeld et al. [23] produced a flexible polyethylene and silicone rubber with a microstructure as the dielectric layer to detect the gravity of flies; Liu et al. [24] developed a porous dielectric layer by adding silver particles in the dielectric layer and improving the sensitivity of the flexible capacitive sensor by 101.5%.

To measure multiple parameters in a confined space, multiple types of composite sensors have been a focus. For example, Luo et al. [25,26] designed a composite inductance and capacitance proximity sensor based on porous anodic alumina (NP-AAO), which has a larger detection distance and detection range. Hyung-kew Lee et al. [27] designed a flexible dual-mode capacitive sensor array that can be used for flexible electronic skin, by connecting different electrode sensor arrays; it can switch between proximity detection and contact detection functions.

In this study, using multiwall carbon nanotubes and polydimethylsiloxane (MWCNTs/PDMS) based on the high-voltage electric field guidance process and the porous foaming process, we designed a new flexible pressure-strain composite sensor to provide signal detection for the control system of fitting the heart surface and adaptive control pressure, which has high sensitivity and repetitive performance while satisfying biocompatibility and compact size.

## 2. Materials and Methods

### 2.1. Exploration on the High-Voltage Electric Field Guidance of MWCNTs

As the force analysis shows in Figure 2, multiwall carbon nanotubes are polarized in the action of electric fields to produce electric dipoles, bringing an electric dipole moment, which can be decomposed orthogonally into component $\alpha_{\|}$ parallel to the carbon nanotubes and component $\alpha_{⊥}$ perpendicular to the carbon nanotubes. The electric dipole is subjected to torque in the external electric field, whose torque size can be expressed as the following formula: (1)$$T=\left|p\wedge E\right|=4\pi \varepsilon_{0}\left(\alpha_{\|}-\alpha_{⊥}\right)E^{2}\sin\theta \cos\theta$$

Formula *T* means torque; *p* means electric dipole moment, vector *E* is electric field strength, unit: V/m; $\varepsilon_{0}$ is a dielectric constant; θ is the angle of distribution direction of carbon nanotubes and the electric field. Due to the special shape of the carbon nanotubes, the polarisation of the parallel carbon nanotubes’ direction is much greater than the vertical direction, so $\alpha_{\|}\gg \alpha_{⊥}$, then the above formula can therefore be reduced to the following form: (2)$$T=\left|p\wedge E\right|=4\pi \varepsilon_{0}\alpha_{\|}E^{2}\sin\theta \cos\theta$$

In a uniformly invariant electric field, the electric dipole moment is already formed when affected by the electric field, so that the torque varies only with the angle. When θ=0, the torque disappears. From the analysis described above, the torque produced by the electric field on the carbon nanotubes causes it to overcome the viscosity drag force of the PDMS, with the final distribution direction consistent with the electric field direction.

If the electric field has not disappeared at this time, due to the existence of electric dipoles, different carbon nanotubes will attract to each other due to the different charge phase of electric dipoles. At this time, the conductivity of the conductive PDMS material will reach its best. The whole procedure is shown in Figure 3.

### 2.2. Preparation of Conductive PDMS Strain Sensor

The preparation process of the conductive PDMS strain sensor was briefly described below:Step 1: prepared the different predetermined weights of PDMS, multiwall carbon nanotubes and appropriate isopropane alcohol in the beaker.Step 2: sonication for 40 min, then heated at 90 °C and isopropanol volatized to obtain an MWCNT-PDMS mixture.Step 3: added the curing agent, then shaped in a v-mold and vacuum treated to remove bubbles.Step 4: heat and cure after the high-voltage electric field treatment (Figure 4a) to obtain the samples of flexible strain sensor (Figure 4b).

### 2.3. Preparation of Porous Conductive PDMS Pressure Sensor

In this study, we used ammonium bicarbonate decomposition to generate pores in the PDMS dielectric layer to prepare a porous PDMS-MWCNT dielectric layer sensor.
Step 1: The conductive PDMS material with 2% carbon nanotubes was prepared and cooled, and 5% ammonium bicarbonate was added.Step 2: After adding the curing agent, the form was heated and cured in the mold. The conductive PDMS porous dielectric layer without adding MWCNTs (Figure 5a) will turn black after adding the MWCNTs.Step 3: The cut into the conductive PDMS material was a 10 mm × 10 mm × 1 mm porous structural dielectric layer with a purple copper electrode plate affixed to its upper and lower surfaces.Step 4: The composite PDMS material was packaged and cured with silicone ecoflex, and the resulting porous PDMS material shown in Figure 5b was the flexible pressure sensor’s finished sample.

## 3. Results

### 3.1. Basic Properties and Micro Characterization of Conductive PDMS Materials

Based on our team’s research on artificial pneumatic hearts, the pneumatic muscle length was 40 mm, the relaxation and contraction were about a 16.5 mm, and the stress variable was 41.25%, so the maximum strain of the conductive PDMS should meet this value. Figure 6a shows the effect of different electric field treatment times on tensile performance at 3% carbon nanotubes concentration. After 20 min of electric field treatment, its tensile strength increased by 62% and its fracture elongation increased by about 18% compared with untreated electric field treatment. The mechanical performance of the test sample was significantly improved.

The sensitivity of the conductive PDMS material was influenced by the formation state of the conductive network, mainly by three factors: carbon nanotube concentration, electric field treatment time, and electric field strength. Figure 6b shows that the peak conductivity PDMS sensitivity of 3% carbon nanotubes was much higher than other concentrations and its fracture elongation was greater than 41.25%; Figure 6c shows the rate of resistance change curves of conductive PDMS materials with different electric field treatment lengths under concentrations of 3% carbon nanotubes. It can be seen that the electric field was treated at 20 min, and the strain range of the high resistance change rate was 40–50%, in line with the longitudinal displacement strain range of the cardiac auxiliary device; Figure 6d shows that with the field strength, the faster the conductive network forms at the same time and the rate of resistance change significantly increases in the conductive PDMS materials.

The morphology of the conductive PDMS material was amplified by randomly taking points with SEM (Figure 7a). It can be seen that on the surface of the electric field guidance directional conductive PDMS material, the carbon nanotubes were evenly arranged, the head and tail were connected to form a local carbon nanotube network, and the direction of the electric field was consistent (Figure 7b). For samples made without electric field guidance, their carbon nanotubes were irregular and more clustered. The electric field can not only improve the arrangement of the carbon nanotubes, but also disperse the clustered carbon nanotubes.

Based on the above experiments, the parameters of conductive PDMS material used in subsequent flexible strain sensors are shown in Table 1 below.

### 3.2. Performance Characterization of Flexible Resistance Strain Sensor

Figure 8 shows the sensitivity characteristics and reciprocating signal lag characteristics of the strain sensor under 0–50% strain, which were described respectively. It can be seen that the sensitivity reaches a peak of 22 under small strain, then stabilizes in the range of 10–15. For hysteresis characteristics, at low strain (less than 20% strain) they were in high signal hysteresis—this may be caused by the displacement of the connecting line; with the increase of deformation (at 30–50% strain), the hysteresis becomes better.

In addition, we also stretch the flexible sensor 50 times with a 50% strain cycle. It can be seen that the peak rate of resistance change of the strain sensor can reach about 55, and its high sensitivity phase stably lasts for 35 cycles, and then the sensitivity becomes low (see Figure 9).

### 3.3. Performance Characterization of Capacitive Pressure Sensor

Experiments show that sensors without carbon nanotubes had low sensitivity, remained around 0.01 Kpa^−1^ and had a peak dielectric constant of less than 2 (Figure 10). Sensors containing carbon nanotubes had capacitance rates of more than 8 in the pressure range of 0–50 Kpa. The rate of capacitance change doubled at 10 Kpa intervals; it was known that its sensitivity was high, all maintained above 0.1 Kpa^−1^. When the pressure was about 280 Kpa, the signal of the flexible pressure sensor tended to be stable, and the pressure range of the sensor detection range fit with a normal heart of 175–283 Kpa [28].

Regarding transient response, a large load was applied to the flexible pressure sensor at 7.2 s, within 180 ms the capacitance was increased from about 20 pf to 100 pf, which was capable of rapidly sensing pressure changes into the pneumatic muscles and responding (Figure 11).

In order to verify the life and stability of the flexible pressure sensor, 300 Kpa cycle pressure was applied to it using a universal test machine, the signal change of 100 times was relatively stable, and the capacitance change rate varied by 1–2 from the end of the initial cycle stage, within the allowable error range (Figure 12).

From the perspective of the overall cycle, except for the cycle time being slightly long in the 1250–1500 s stage, the overall changes were relatively uniform. The flexible pressure sensor prepared at this institute had a relatively long life and stability. These advantages were required for the application of the cardiac auxiliary device.

### 3.4. Composite and Performance of Flexible Sensors

The flexible resistance strain sensor and flexible capacitive pressure sensor had the advantages of high sensitivity and good repetition, but there is no effective combination method for the application of a direct cardiac auxiliary device, and it is necessary to combine the actual installation position to solve many problems such as human safety and space restrictions. In order to detect the force and displacement of the pneumatic muscles and to be close to the application surface of the muscles, the composite sensor should be installed between the heart and the pneumatic muscles. The strain and pressure sensors should be composite, as shown in Figure 13. We used a model heart to test the capacitance value and the pressure of the pneumatic muscle (see Figure 14).

The three cycles of the pressure were 0–20 Kpa, the pressure signal of the composite sensor was also circulating, and the amplitude of each change was relatively stable. When the air pressure input was 15 Kpa, the capacitive value was mutated when the surface pneumatic muscles fit the heart (Figure 15).

Air pressure values will affect the deformation of the pneumatic muscles and change the strain value measured by the composite sensor. Performing three times in the range of 18–22 Kpa, air pressure circulation of the ventricular assist device measured the resistance changes of the strain (Figure 16).

The results showed that the signal of the sensor also had three stable cycles, and was dense at a resistance of 2100 KΩ, with an 18 Kpa normal operating pressure. The maximum resistance value was about 240 KΩ, which corresponds to an input pressure of 22 Kpa. It can be explained that the composite sensor can capture the resistance value corresponding to the critical working air pressure, and can complete the air pressure control and overpressure protection of the ventricular assist device.

## 4. Discussion

The composite flexible stress-strain sensor proposed in this paper could better adapt to the difference of individual heart sizes and the problem of air pressure control of direct ventricular assist devices. Based on experiments, it was shown that this composite sensor had a longer high sensitivity repeat compared to those in previous studies.

However, 35 cycles of strain application was far from enough, so a longer cycle sensor design still needs further discussion. In addition, the research on the sensitive elements with different geometries in flexible sensors has great potential. The sensitivity and stability of sensors can be improved more effectively by manufacturing more complex grid or three-dimensional array sensitive elements.

## Figures and Tables

**Figure 1 sensors-22-02607-f001:**
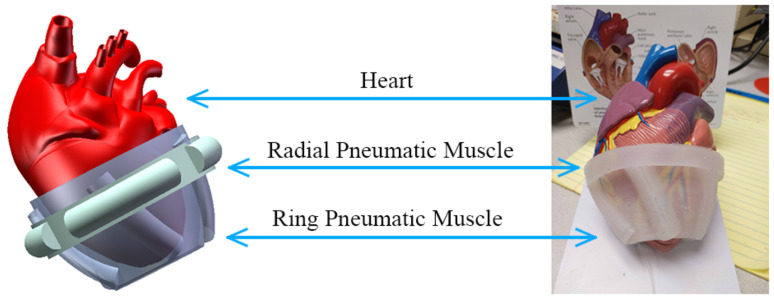
The direct ventricular assist device.

**Figure 2 sensors-22-02607-f002:**
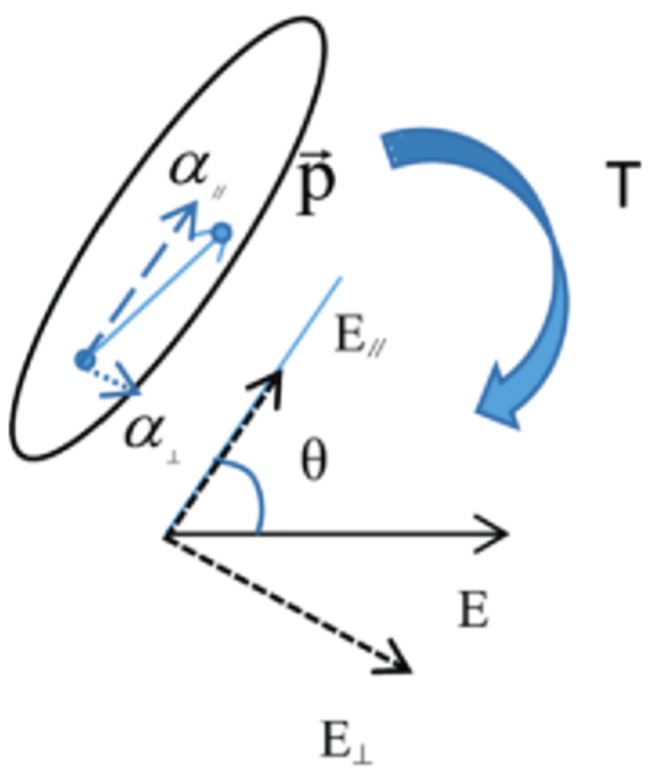
Electric-field force analysis of MWCNT.

**Figure 3 sensors-22-02607-f003:**
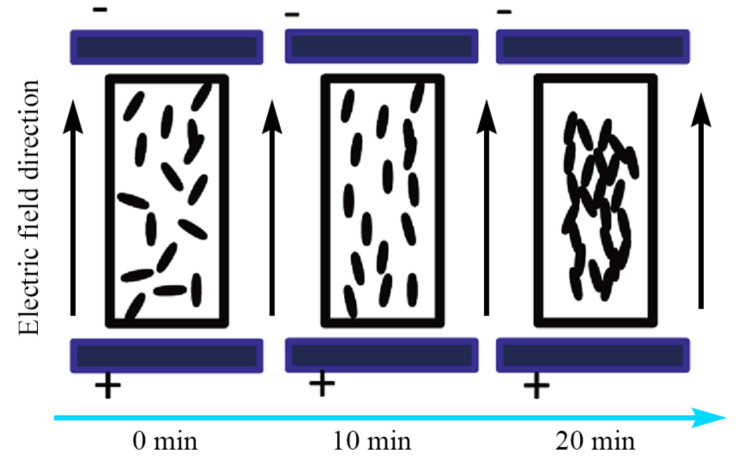
Affect of the electric field on the MWCNTs in PDMS over time.

**Figure 4 sensors-22-02607-f004:**
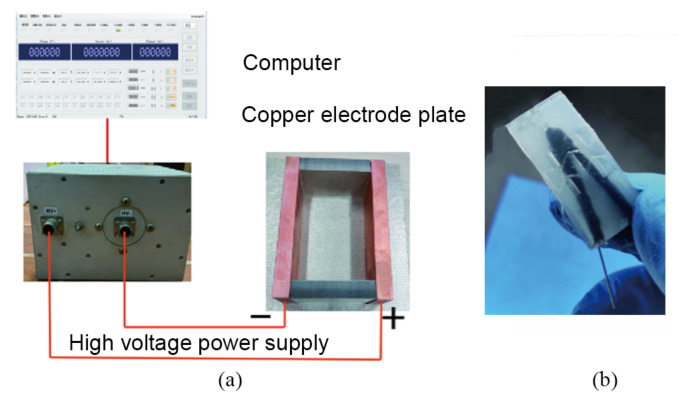
(**a**) Electric field treatment environment. (**b**) Conductive PDMS strain sensor.

**Figure 5 sensors-22-02607-f005:**
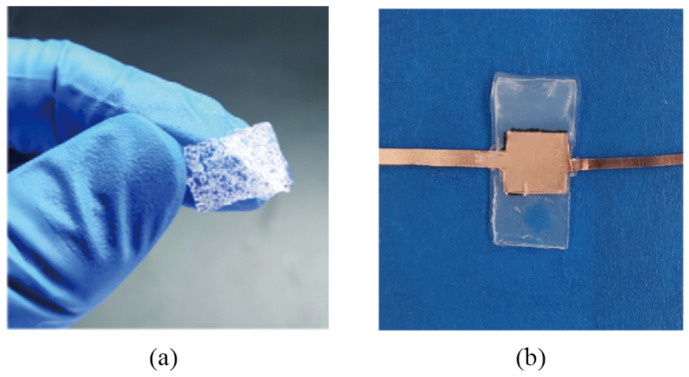
(**a**) The resulting porous PDMS material without MWCNTs. (**b**) The capacitive flexible pressure sensor.

**Figure 6 sensors-22-02607-f006:**
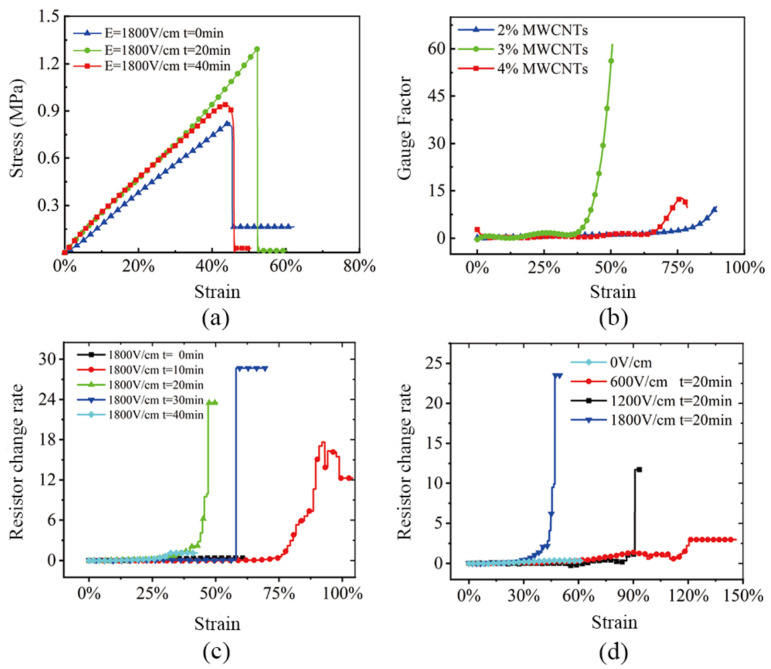
(**a**) Different electric field treatment times for tensile performance. (**b**) Effect of the carbon nanotubes concentration on the sensitivity. (**c**) Effect of treatment time on the rate of resistance. (**d**) Effect of the electric field strength on the sensitivity.

**Figure 7 sensors-22-02607-f007:**
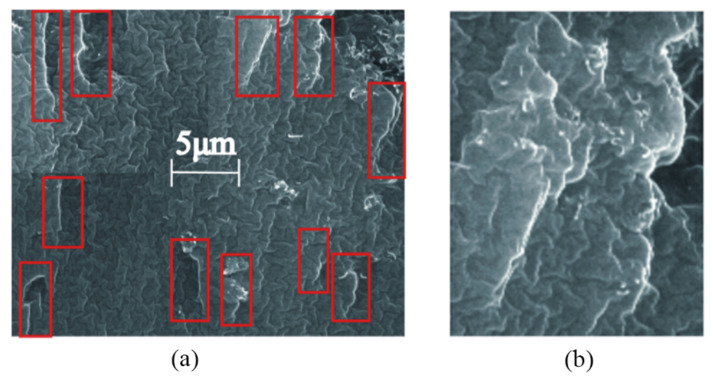
(**a**) SEM photograph of conductive PDMS material. (**b**) The local carbon nanotubes network.

**Figure 8 sensors-22-02607-f008:**
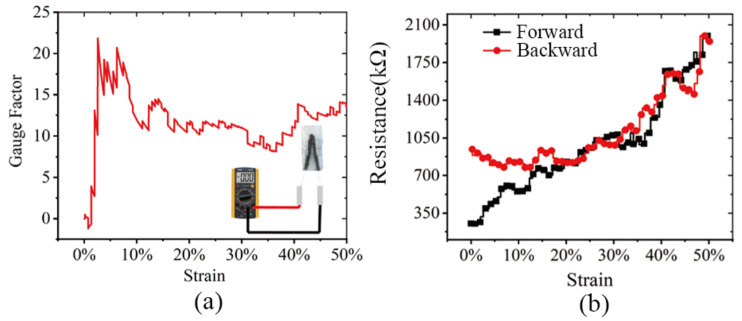
(**a**) The sensitivity characteristics. (**b**) The late hysteresis characteristics.

**Figure 9 sensors-22-02607-f009:**
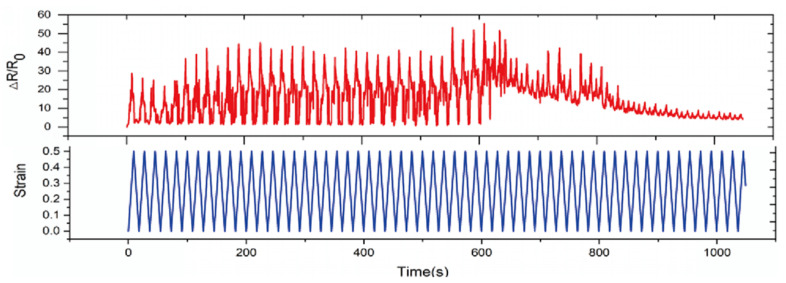
The repeatability of the strain sensor.

**Figure 10 sensors-22-02607-f010:**
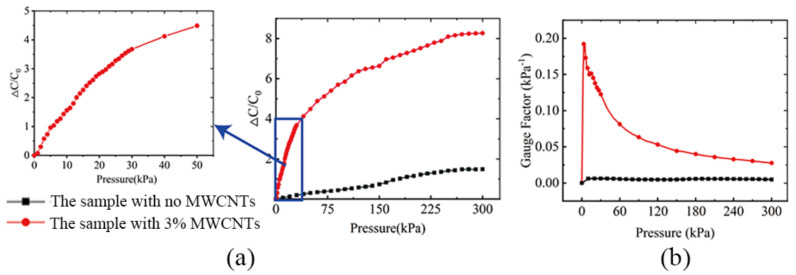
(**a**) Comparison diagram of the capacitance change rate of the flexible pressure sensor. (**b**) Comparison diagram of the sensitivity of the flexible pressure sensor.

**Figure 11 sensors-22-02607-f011:**
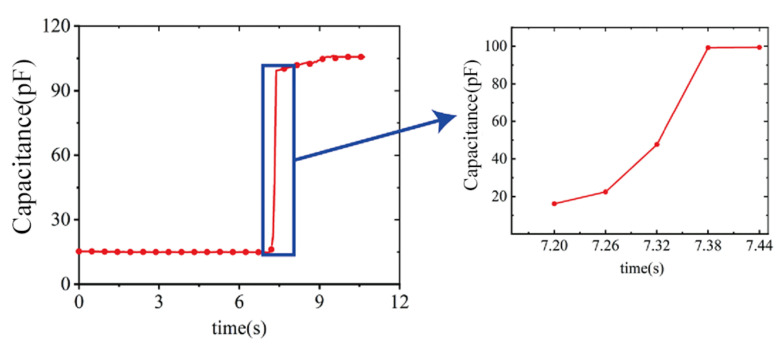
Transient response of the flexible pressure sensor.

**Figure 12 sensors-22-02607-f012:**
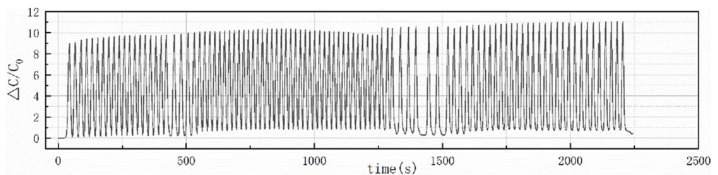
Repeatability of the flexible pressure sensor.

**Figure 13 sensors-22-02607-f013:**
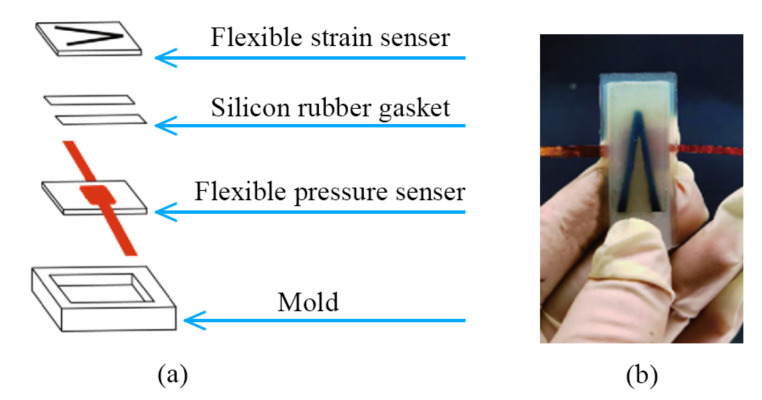
(**a**) Structure of the composite sensor. (**b**) The actual composite sensor.

**Figure 14 sensors-22-02607-f014:**
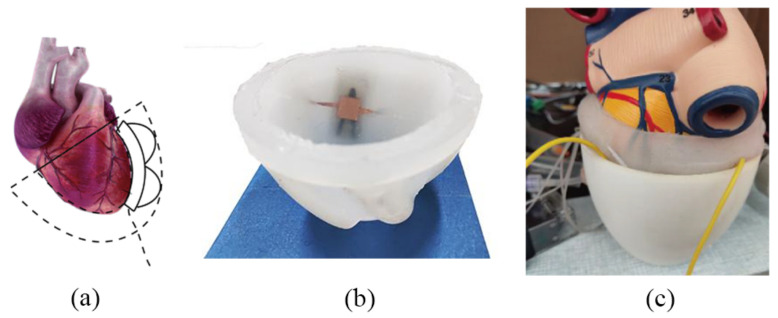
(**a**) Installation position of the pneumatic muscles. (**b**) Installation position of the composite sensor. (**c**) Experiments were performed with the model heart.

**Figure 15 sensors-22-02607-f015:**
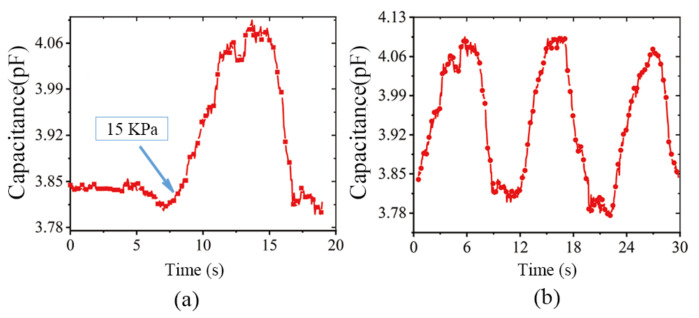
(**a**) The point of mutation at 15 Kpa. (**b**) Three beating cycles test.

**Figure 16 sensors-22-02607-f016:**
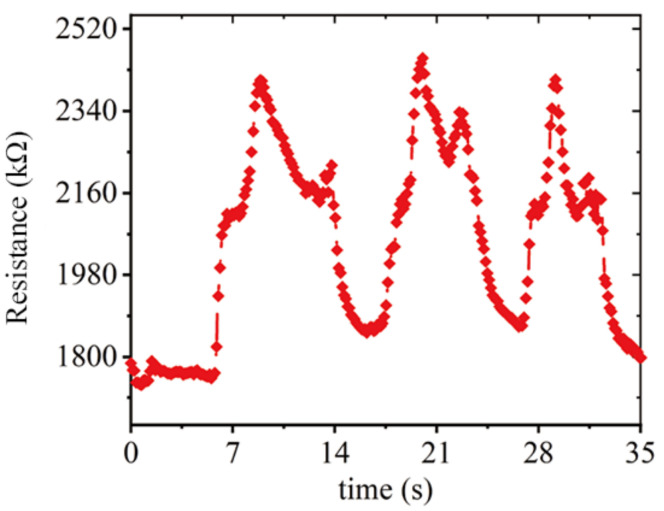
Resistance signal diagram of air pressure control experiment.

**Table 1 sensors-22-02607-t001:** Preparation parameters of conducting medium.

MWCNTs Concentration (%)	Electric Field Treatment Duration (min)	Electric Field Intensity (Volt/cm)
3	20	1800

## Data Availability

Not applicable.

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
