# Peer review of "A Composite Flexible Sensor for Direct Ventricular Assist Device"

_sensors, 2022, doi:10.3390/s22072607_

Round 1

Reviewer 1 Report

I would suggest commenting better on the figures, in general.

Author Response

Thank you for your comments!

Reviewer 2 Report

The authors reported the development of a composite flexible sensor for direct ventricular assist device in this manuscript. The research idea is greatly achieved throughout sequential experimental steps and complex discussion in the main text. However, there are some lacks that must be addressed to significantly improve the draft arrangement, writing, and data presentation of this work. Here are some comments, which would be helpful to guide the revision process:

Major comments:

  1. Please refer to the journal criteria when preparing the whole research manuscript since it is quite critical for the basic screening of any draft. The authors did not separate the materials & methods, results, and discussion in the current manuscript version, Instead, all of them are written together. Thus, it is highly recommended to rearrange the scientific writing carefully.
  2. In some specific sections, there are too much information and descriptive explanation, which makes this manuscript looks like a review, while this is going to be a research article. Please try to reduce the unimportant statements or try to summarize the previous underlying studies that support the current findings, so that they would be concise.
  3. Please try to manage the figure contents since there are too many main figures in the manuscript. The reviewer personally thinks that some figures could be moved as supplementary information. Moreover, some figures might be better to be combined into an integrative figure, which might be helpful to reduce the number of figures as well. Please have a concern to only the critical, impactful, or interesting data should be put as the main figures. please consider the interest of the readership when the paper is way too long and complicated.
  4. In section 2.3, what is the fundamental that underlies the initial assumption value of pneumatic muscle length, relaxation and contraction, and stress variable? Please provide any literature or facts about this issue if possible.
  5. Regarding the perspective for basic principles of sensors, how could the authors carry out the specificity factor of this composite flexible sensor model? Some additional explanations might signify the assurance of the proposed sensor.
  6. Since this sensor device is strongly related to a human organ orientation, is there any preliminary study on the in vivo model using the proposed sensor? It could be a significant aspect that will be worth the efforts.
  7. In the case of the flexible resistance strain sensors, there are two types of molds used for the sensor fabrication (type I and V). Are there any specific reasons for choosing those shapes? By any chance, are there any other shapes also applicable?
  8. Based on the results in Figure 9, it is asserted that the type V sensor arrangement has better sensitivity. Could the authors deeply explain the scientific rationale of that phenomenon?
  9. The contents in sections 4.1 and 4.2 are switched, which this error could be very critical. Following on Figure 18 in section 4.1, it belongs to the pressure sensor, not the strain sensor. This is also vice versa for section 4.2. Please take care of this issue carefully.

Minor comments:

  1. The X- and Y-axis information was missed in Figure 11.
  2. On page 2 line 58, this ‘(P(VDF-TrFe)’ word might be mistyped. Please check it.
  3. Many grammatical errors were found: page 4 line 130 (vacacuum should be vacuum); throughout the whole manuscript (ware should be were); page 8, line 210 (capacacitive should be capacitive); page 9 line 236 (capacittive should be capacitive)
  4. Many spacing errors and wrong punctuation marks were found. Please carefully check the whole manuscript.
  5. Please pay attention to the writing format of any specific unit, whether it should be in the superscript or subscript form. Many wrong unit formats were found.

Author Response

Thank you for your comments!

Reviewer 3 Report

This manuscript presented the development of direct ventricular assist device using MWCNT and PDMS composites. The topic is interesting and important. However, the authors need to more clarify both strain and pressure sensors as a ventricular failure device. Therefore, I cannot recommend for the publication of its present form in this high impact journal.

Comments:

  1. During molding of composites, vacuum was conducted for removing bubbles; however, I believe bubbles cannot completely remove. How bubble remove after vacuum of composites.
  2. As we know when high voltage applied to polymer, breakdown happens at a certain voltage, Authors believe it increase conductivity. What is the benefit of such treatment for those sensors. I could not see any improvement of the strain sensors, since many MWCNTs based polymer showed more than 70% strain.
  3. Connecting wire is important for strain sensors, and it is easy to displace during stress applied on the sensor. How authors improve this issue, although authors made sensors using sandwiched type to improve the strain of flexible composites.
  4. Reproducibility is important for sensors. I did not find reproducibility test in the work.
  5. As compared to Fig. 8 a and Fig. 23, why resistance started to increase 1800 ohm? Authors need to clarify in the revised manuscript.
  6. What are the advantages this proposed work over other published works particular ventricular assist device.
  7. Abstract and conclusion should be sound and concise.

Author Response

Thank you for your comments!

Round 2

Reviewer 2 Report

The revised manuscript version is quite improved and thus it can be recommended to publish it accordingly.

Reviewer 3 Report

I beleive authors carefully revised the comments  and reflected in the revised manuscript. I therefore  recommended to publish the manuscript in Sensors Journal.